**Data Availability Statement:** Dataset was deposited on 10.6084/m9.figshare.24872052.

**Funding:** The authors received no specific funding for this work.

# Characteristics of consecutive versus non-consecutive frequent emergency medical services transport to a single emergency department

**Sun Hyu Kim** ⬡ *, **Hyeji Lee**

Department of Emergency Medicine, University of Ulsan College of Medicine, Ulsan University Hospital, Ulsan, Republic of Korea

* stachy1@paran.com

## Abstract

### Objective

This study was to examine characteristics concerning frequent users of emergency medical services (EMS) transport by comparing patients who used EMS transport frequently for one year and those who used EMS transport for more than two years consecutively.

### Methods

A retrospective review for frequent use of EMS transport was conducted. The patients from the fire stations that transported more than 70% of all EMS transport to the study hospital emergency department (ED) were included. The study subjects were divided into consecutive group (frequent EMS transport for ≥ two years consecutively) and non-consecutive group (frequent EMS transport for only one year). Characteristics of patients who were frequent users of EMS transport and those of all cases with EMS transport were examined.

### Results

Of the total 205 patients and 1204 cases of frequent EMS transport, 85 (42%) patients and 755 (63%) cases were in the consecutive group. Patients in the consecutive group were more likely to have risky alcohol use, unemployed state, and medical aid type of payment for ED treatment than those in the non-consecutive group. More patients had previous experience of EMS transport to the study hospital ED in the consecutive group and the number of cases with alcohol ingestion was higher in the consecutive group. Elapsed time from EMS call to ED arrival was longer for the consecutive group.

### Conclusion

Risky alcohol use, unemployed state, and previous experience of EMS transport were associated with consecutive and frequent use of EMS transport in frequent users of EMS transport.

**Competing interests:** The authors have declared that no competing interests exist.

## Introduction

Transport via Emergency medical services (EMS), also known as public ambulance services, accounted for 13%~35% of all emergency department (ED) arrivals [1–3]. It has been reported that 67~90% of calls for EMS transport are actually transferred to ED [4, 5]. More than 10% of users of EMS transport are not emergent or necessary to obtain immediate medical intervention [6, 7]. Life-threatening diagnoses are only made for 20% of EMS transport users. Therefore, many EMS users are not emergent in nature [8]. EMS transport should be provided to emergent patients only. If it is not used appropriately, many emergent patients will not have access to EMS transport.

ED visitors who frequently use EMS transport are also increasing [5, 9, 10]. Those who have substance abuse, mental illness, seizure, and respiratory diseases are frequent users of EMS transport [11, 12]. Many of these frequent users of EMS transport are covered by medical aid, a type of medical payment system in Korea [5, 13, 14]. Some characteristics such as previous experience of using EMS by patients might be associated with frequent use of EMS transport. Such patients tend to have higher understanding of EMS transport operation. Therefore, they are more likely to call for EMS transport.

Cases involving uncooperative patients, the denial of ED treatment after EMS transport, the absence of any specific symptoms at ED arrival, and repetitive use with the same symptoms might all be defined as instances of unnecessary EMS transport, along with other cases [15]. Although the judgment of unnecessary EMS transport might later turn out to be erroneous, it is important to reduce unnecessary EMS transport to make EMS more efficient. To this end, it is important to conduct accurate analysis of frequent use of EMS transport, which is likely to account for a large portion of unnecessary EMS transport. Frequent users of EMS transport have been studied based on EMS department data, not hospital data in the previous studies [11, 12, 16, 17]. Using data based on EMS department data without hospital data was limited in its ability to identify the characteristics of frequent users of EMS transport in further detail. Moreover, within frequent users of EMS transport, the characteristics of less or more frequent users of EMS transport might differ. Therefore, the objective of this study was to examine characteristics concerning frequent users of EMS transport by comparing patients who used EMS transport frequently for one year and those who used EMS transport for more than two years consecutively.

## Methods

### Data source and subjects

Patients ≥ 16 years old who were presented to the university training hospital ED located in southeast coast area of South Korea between January 2011 and December 2015 were included in this study. The anonymous data were analyzed retrospectively. Data was accessed on 4th September 2017. Informed consent was exempted due to retrospective study and this study was approved by the institutional review board.

The definition of frequent use of EMS transport was arbitrary following previous studies [18, 19]. In this study, frequent calls for EMS transport were considered if patients called for EMS transport for more than five times per year [8]. However, not all patients who called for EMS transport were transferred to ED. In Korea, it has been reported that 67% of patients calling for EMS transport are transferred to ED [4]. Therefore, we defined frequent users of EMS transport as those who visited ED using EMS transport for more than three times per year. To evaluate characteristics of frequent users of EMS transport, we classified frequent users of EMS transport into two groups: 1) those who used EMS transport frequently for only one year

(non-consecutive group); and 2) those who used EMS transport frequently for more than two years consecutively (consecutive group). Consecutive group were defined as patients who visited ED using EMS transport more than three times for one year and more than five times for two years consecutively [20]. While there is a possibility of being classified into the consecutive group depending on the extension of the study period or the time of death, frequent users of EMS transport for only one year during the study period were defined as a non-consecutive group. For example, to be classified cases of the frequent user of EMS transport for the first time in the last year of the study period, death in the first year of the study period, or moving to a distant area from the study hospital in the first year of the study period were defined as a non-consecutive group even if there was a possibility of being classified into the consecutive group if the study period was extended or the time of death was after the second year of the study period.

## Setting

The study hospital is located in a highly industrialized city with 1,200,000 residents. Fire department is responsible for prehospital EMS treatment and transport in South Korea. Prehospital EMS personnel are mostly registered nurses or emergency medical technicians (EMTs) at the intermediate level, and EMTs at the intermediate level must graduate from a department for EMT training for 3 or 4 years. In principle, prehospital EMS personnel should transport the patients to the hospital after they are dispatched to the scene and medical direction from a medical director mainly an emergency physician was necessary in order not to transport the patient. Twenty-seven fire stations operate prehospital public ambulance for emergent patients in this city during the study period. Fire stations of the city are grouped into four divisions (east, central, south, and others) based on their geographic locations. There are ten EDs, including two training hospital EDs in the city. The study hospital is the biggest one which is located at east shore area of the city. Four fire stations are in charge of EMS transport for the east area. Because the study hospital ED is the only one in the east area of the city, most patients of the east area are transported to the study hospital ED by the four fire stations. Three of these four fire stations transport more than 90% of patients to the study hospital ED and one of these four fire stations transports more than 70% of patients to the study hospital ED (dominant fire station). Although fire stations located in other divisions of the city also transport patients to the study hospital ED, their transport accounts for less than 30% of all EMS transport (non-dominant fire stations). If patients transported to the study hospital ED from non-dominant fire stations are included in this study, we could not analyze data of most patients transported to other hospital EDs. Therefore, we only included patients transported by dominant fire stations to determine the frequent use of EMS transport more accurately (Figs 1 and 2).

## Data collection and variables

Characteristics of patients who were frequent users of EMS transport and those of all cases with EMS transport were examined (Fig 2). Hospital data and EMS run sheet written by EMS personnel were evaluated. EMS run sheet included fire station, date, time of call for EMS transport, time of arrival to patient, time of arrival to ED, occupation, place of call for EMS transport, alcohol ingestion, symptoms, and mental status. Place of call for EMS transport was classified into home, residential area other than home, transport area, and others. Symptoms were divided into non-disease related symptoms (including trauma, intoxicated cases, and so on) and disease related symptoms. Type of medical payment for ED treatment, experience of EMS transport to the study hospital ED before the study period, and the ratio of EMS transport

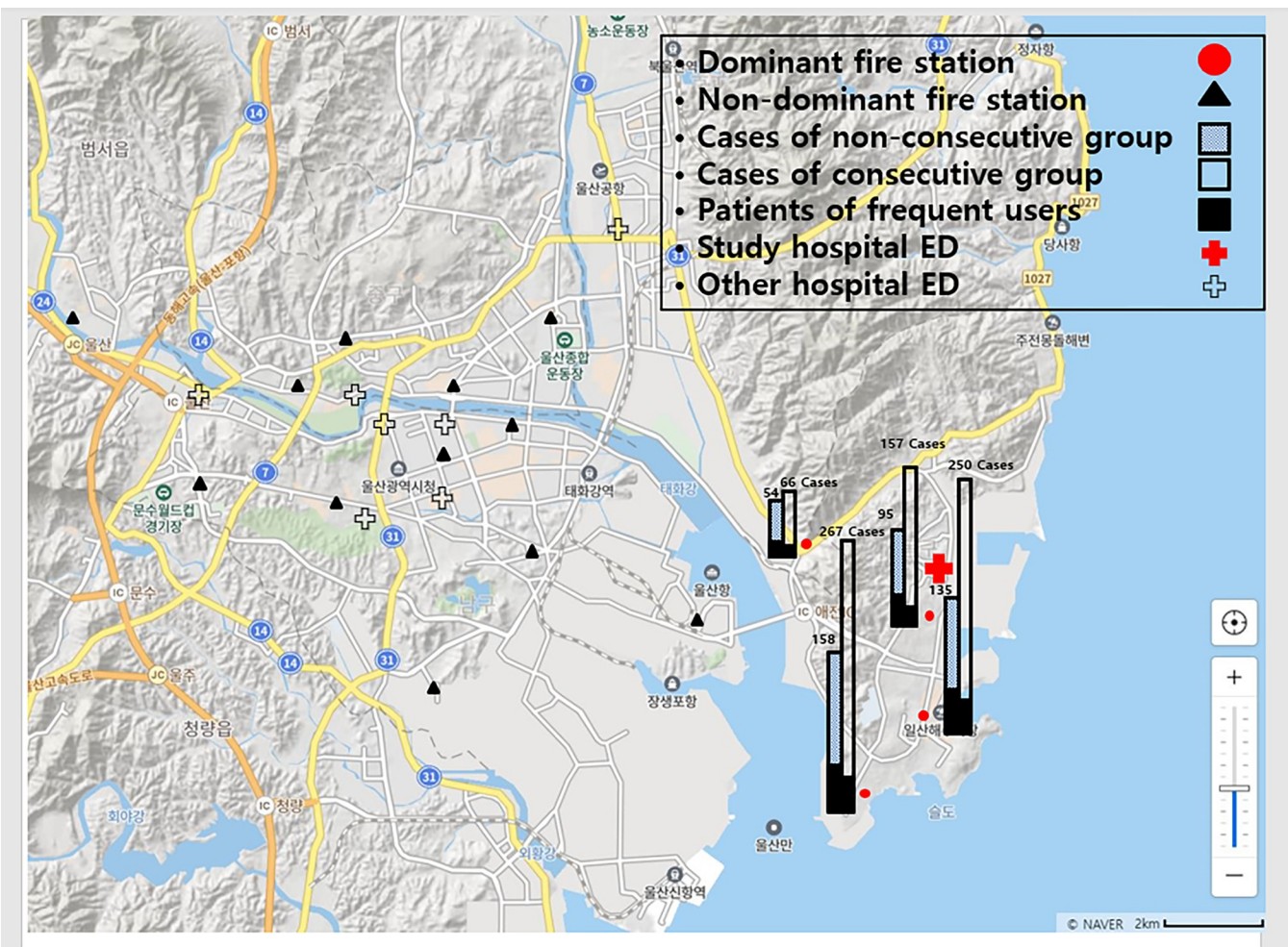

**Fig 1. Dominant fire stations transported more than 70% of all emergency medical services (EMS) transport to the study hospital emergency department (ED) while non-dominant fire stations transported less than 30% of all EMS transport to the study hospital ED.** The number of cases transported to the study hospital ED from dominant fire stations is more in the consecutive group (white bar) than that in the non-consecutive group (multiple dotted bar), although the number of patients (black bar) is less in the consecutive group. Two other hospital EDs and 9 non-dominant fire stations are located outside the map.

with the same purpose or symptoms in all frequent uses of EMS transport were investigated. Age was recorded as the age at the time of EMS transport and divided into two group: 16 ~ 64 years old and ≥ 65 years old. Time of call for EMS transport or ED arrival was divided by 6 hours. Seasons were also examined to capture any differences in frequent EMS transport depending on the season by classifying months into the seasons of spring (March-May), summer (June-August), autumn (September-November), and winter (December-February). Mental status was divided into alert or not-alert mentality. Previous drinking behavior was defined using guidelines of National Institute on Alcohol Abuse and Alcoholism [21]. A standard drink is any drink that contains about 14 grams of pure ethanol. Beer and Soju are famous liquors in Korea. A can (355 mL) of beer containing 5% of alcohol can be a standard drink. A glass (80 mL) of Soju containing 18% of alcohol can also be a standard drink. Risky alcohol use is considered when men or adult < 65 years old drink more than 14 standard drinks per week on average or more than 4 drinks on any day or when women or adults ≥ 65 years old years drink more than 7 standard drinks per week on average or more than 3 drinks on any day.

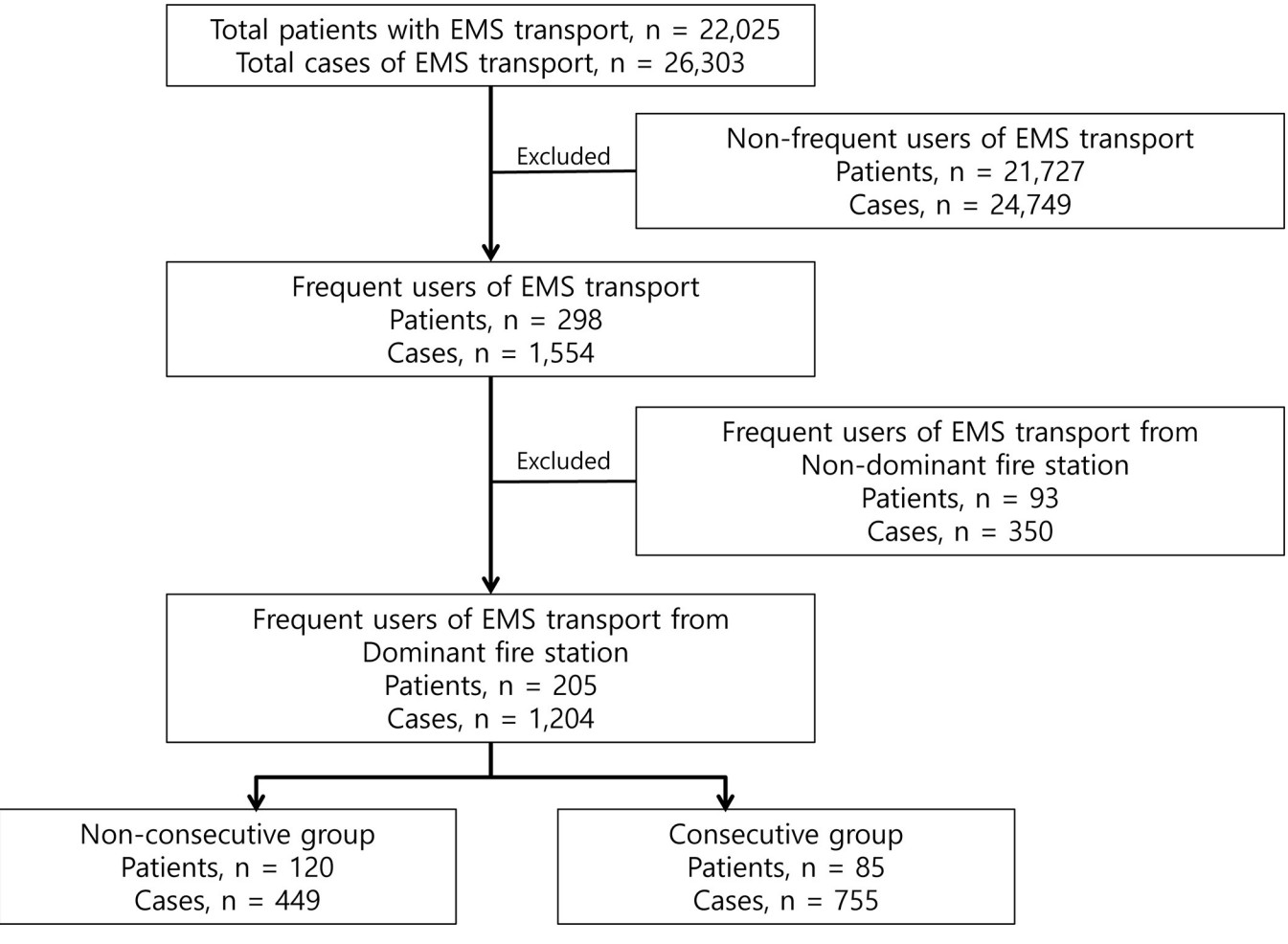

**Fig 2. Flow chart showing the selection of study subjects.**

Non-risky alcohol use is considered when one drinks less than amount of risky alcohol use or does not drink at all. The worst drinking behavior was selected as a representative one with a change in drinking behavior. Results of ED treatment included normal discharge from ED, admission to general ward or intensive care unit (ICU), and noncooperation to ED treatment. Patients who refused any treatment at the ED and discharged with their own will regardless of medical advice or escaped from the ED were determined as noncooperation to ED treatment. Occupation was divided into employed or unemployed state. Students and housewives were considered as employed. Type of medical payment for ED treatment was divided into non-medical aid and medical aid. Medial aid was considered if patients had financial support from the government (Korean government or local government) for ED treatment.

## Statistical analysis

Chi-square, Fisher's exact test, and Student's *t*-test were used to compare the non-consecutive group and the consecutive group. Each analysis was performed for patients and those of all cases with EMS transport in both groups. Multivariate logistic regression with forward step-wise method was performed using significant variables at $p < 0.05$ from univariate comparison to determine significant factors associated with more consecutive use of EMS transport in

frequent users. IBM SPSS 21.0 (IBM, Armonk, NY, USA) was used for all statistical analyses. Statistical significance was set at $p < 0.05$.

## Results

There were a total of 1204 cases of frequent EMS transport, including 205 patients with frequent EMS transport. Eighty-five (42%) of these 205 patients and 755 (63%) of 1024 cases were in the consecutive group (Fig 2). The median number (interquartile range) of EMS transport of the consecutive and non-consecutive groups were 7 (6–9) and 4 (3–4) respectively.

Male to female ratio or percentage of patients with past medical history was not significantly different between the consecutive group and the non-consecutive group. Patients in the consecutive group were more likely to have risky alcohol use, unemployed state, and medical aid type of payment for ED treatment than those in the non-consecutive group. The percentage of patients with previous experience of EMS transport to the study hospital ED was 60% in the consecutive group and 35% in the non-consecutive group. Mortality during or after the study period was higher in the consecutive group than that in the non-consecutive group (Table 1).

There was no significant difference in age group, season, time or day of call for all cases of EMS transport between the two groups. The number of cases with alcohol ingestion was higher in the consecutive group than that in the non-consecutive group. The most common place of call for EMS transport was at home in both groups (71% in the non-consecutive group and 84% in the consecutive group). Elapsed time from EMS call to ED arrival was longer for the consecutive group than that for the non-consecutive group (Table 2).

The most common symptom was abdominal pain in the consecutive group and other pain in the non-consecutive group (Table 3).

Table 1. General characteristics of patients with frequent EMS transport in the non-consecutive group and the consecutive group.

| | Non-consecutive (n = 120) | Consecutive (n = 85) | p-value |
|---|---|---|---|
| Male sex, n (%) | 67 (55.8) | 48 (56.5) | 0.928 |
| Risky alcohol use, n (%) | 33 (27.5) | 37 (43.5) | 0.017 |
| Unemployed state, n (%) | 62 (51.7) | 65 (76.5) | <0.001 |
| Medical history, n (%) | | | |
| Psychiatric history | 29 (24.2) | 25 (29.4) | 0.401 |
| Hypertension | 41 (34.2) | 40 (47.1) | 0.063 |
| Diabetes Mellitus | 35 (29.2) | 28 (32.9) | 0.564 |
| Cerebrovascular disease | 35 (29.2) | 28 (32.9) | 0.564 |
| Cardiovascular disease | 26 (21.7) | 22 (25.9) | 0.483 |
| Pulmonary disease | 22 (18.3) | 22 (25.9) | 0.195 |
| Liver disease | 31 (25.8) | 32 (37.6) | 0.071 |
| Renal disease | 20 (16.7) | 17 (20.0) | 0.541 |
| Malignancy | 15 (12.5) | 17 (20.0) | 0.145 |
| Previous experience of EMS transport, n (%) | 42 (35.0) | 51 (60.0) | <0.001 |
| Same purpose or symptoms in all frequent    EMS transport, % | 77.4 ± 23.9 | 79.3 ± 21.8 | 0.573 |
| Health insurance system, n (%) | | | 0.003 |
| Non-medical aid | 90 (75.0) | 47 (55.3) | |
| Medical aid | 30 (25.0) | 38 (44.7) | |
| Death, n (%) | 32 (26.7) | 37 (43.5) | 0.012 |

EMS: emergency medical services.

**Table 2. General characteristics of total cases of frequent EMS transport in the non-consecutive group and the consecutive group.**

| | Non-consecutive (n = 449) | Consecutive (n = 755) | p-value |
|---|---|---|---|
| Age group, n (%) | 52 (38–67)* | 56 (48–67)* | 0.119 |
| 16~64 years old | 326 (72.6) | 516 (68.3) | |
| ≥65 years old | 123 (27.4) | 239 (31.7) | |
| Season of EMS transport, n (%) | | | 0.051 |
| Spring | 139 (31.0) | 187 (24.8) | |
| Summer | 114 (25.4) | 180 (23.8) | |
| Autumn | 96 (21.4) | 186 (24.6) | |
| Winter | 100 (22.3) | 202 (26.8) | |
| Time of EMS transport, n (%) | | | 0.802 |
| ≤00 ~ <06 | 95 (21.2) | 144 (19.1) | |
| ≤06 ~ <12 | 108 (23.6) | 181 (24.0) | |
| ≤12 ~ <18 | 124 (27.6) | 223 (29.5) | |
| ≤18 ~ <24 | 124 (27.6) | 207 (27.4) | |
| A day of EMS transport, n (%) | | | 0.417 |
| Weekdays | 307 (68.4) | 533 (70.6) | |
| Weekends and holidays | 142 (31.6) | 222 (29.4) | |
| Alcohol ingestion, n (%) | 82 (18.3) | 240 (31.8) | <0.001 |
| Place of call for EMS transport, n (%) | | | <0.001 |
| Home | 320 (71.3) | 635 (84.1) | |
| Residential area other than home | 19 (4.2) | 41 (5.4) | |
| Transportation area | 65 (14.5) | 32 (4.2) | |
| Others | 45 (10.0) | 47 (6.2) | |
| Elapsed time from EMS call to, min | | | |
| Scene | 5.0±2.1 | 5.0±2.4 | 0.906 |
| Arrival at ED | 18.8±6.6 | 20.0±7.6 | 0.005 |

* Median age (interquartile range)

EMS: emergency medical services; ED: emergency department.

Regarding characteristics of symptoms, 73% patients in the non-consecutive group and 82% patients in the consecutive group showed disease related symptoms. Noncooperation to ED treatment after ED arrival occurred in 6% patients in the non-consecutive group and 16% in the consecutive group (Table 4).

Results of multivariate logistic regression conducted using meaningful variables in univariate comparison in Table 1, showed that consecutive and frequent use of EMS transport was related to risky alcohol use, un-employed state, and previous experience of EMS transport in frequent users of EMS transport (Table 5).

## Discussion

In this study, patients with risky alcohol use, unemployed state, and previous experience of EMS transport were found to use EMS more consecutively among the frequent EMS users. Analysis of frequent EMS use is necessary to identify inappropriate EMS use, but frequent EMS use should not be equated with inappropriate EMS use. Many of the frequent EMS uses are appropriate EMS use, and it will be necessary to accurately analyze inappropriate EMS use and appropriate EMS use among frequent EMS uses.

The national health insurance system covers all people in South Korea. Individual insurance premiums for national health insurance are different according to their annual incomes.

**Table 3. Symptoms of total cases of frequent EMS transport in the non-consecutive group and the consecutive group.**

| | Non-consecutive (n = 449) | Consecutive (n = 755) | p-value |
|---|---|---|---|
| Symptom of patients, n (%) | | | |
| Headache | 14 (3.1) | 25 (3.3) | 0.855 |
| Chest pain | 20 (4.5) | 22 (2.9) | 0.159 |
| Abdominal pain | 61 (13.6) | 130 (17.2) | 0.095 |
| Back pain | 3 (0.7) | 23 (3.0) | 0.006 |
| Other pain | 105 (23.4) | 129 (17.1) | 0.008 |
| Laceration/abrasion | 8 (1.8) | 15 (2.0) | 0.802 |
| Mental deterioration | 45 (10.0) | 80 (10.6) | 0.752 |
| Dizziness | 3 (0.7) | 16 (2.1) | 0.051 |
| Vertigo | 7 (1.6) | 3 (0.4) | 0.046 |
| Extremity weakness | 4 (0.9) | 8 (1.1) | 1.000 |
| General weakness | 21 (4.7) | 38 (5.0) | 0.782 |
| Syncope | 24 (5.3) | 10 (1.3) | 0.000 |
| Respiratory difficulty | 41 (9.1) | 98 (13.0) | 0.043 |
| Cardiac arrest | 5 (1.1) | 7 (0.9) | 0.770 |
| Vomiting/diarrhea/constipation | 10 (2.2) | 32 (4.2) | 0.066 |
| Hematemesis | 4 (0.9) | 16 (2.1) | 0.107 |
| Fever | 13 (2.9) | 15 (2.0) | 0.312 |
| Psychiatric disorder | 2 (0.4) | 9 (1.2) | 0.227 |
| Others | 52 (11.6) | 69 (9.1) | 0.173 |

**Table 4. Clinical characteristics of total cases of frequent EMS transport in the non-consecutive group and the consecutive group.**

| | Non-consecutive (n = 449) | Consecutive (n = 755) | p-value |
|---|---|---|---|
| Characteristics of symptoms, n (%) | | | <0.001 |
| Disease | 327 (72.8) | 620 (82.1) | |
| Non-disease | 122 (27.2) | 135 (17.9) | |
| Not-alert mental status, n (%) | 71 (15.8) | 116 (15.4) | 0.835 |
| Results of ED treatment, n (%) | | | <0.001 |
| Discharge | 309 (68.8) | 442 (58.5) | |
| Admission | 114 (25.4) | 193 (25.6) | |
| Noncooperation to ED treatment | 26 (5.8) | 120 (15.9) | |
| Systolic blood pressure, mmHg | 132.5 ± 27.3 | 132.6 ± 28.4 | 0.953 |
| Diastolic blood pressure, mmHg | 80.5 ± 17.5 | 80.9 ± 18.1 | 0.706 |

ED: emergency department.

**Table 5. Factors associated with more consecutive use of EMS transport in frequent users of EMS transport.**

| | Odds Ratio | 95% CI | p |
|---|---|---|---|
| Risky alcohol use versus non-risky alcohol use | 2.004 | 1.071–3.749 | 0.030 |
| Unemployed state versus employed state | 2.823 | 1.488–5.356 | 0.001 |
| Previous experience of EMS transport | 2.351 | 1.292–4.276 | 0.005 |

EMS: emergency medical services.

However, insurance benefits are not different. Medical aid program is a part of public assistance. It is available to patients who do not have enough to pay individual insurance premiums for national health insurance. Approximately 2.97% of all South Koreans and 1.52% of all citizens of this city are recipients of medical aid [22]. EMS transport is provided free of charge to all people regardless of their type or possession of national health insurance in South Korea. However, EMS transport is not provided free of charge to all people in some countries. In the United States, patients with Medicare type of payment have to pay for EMS transport in most cases. However, patients with Medicaid type of payment are free of charge for EMS transport in some emergent cases. Twenty-one percent of all EMS users had Medicaid type while 53% of all frequent EMS users had Medicaid type. Despite the possibility of EMS transport cost, patients with Medicaid type are more likely to use EMS transport [5]. In the present study, 25% of patients in the non-consecutive group and 45% of patients in the consecutive group had medical aid type of payment. This result shows that patients with medical aid type of payment tend to use EMS transport more frequently.

In a previous study about unnecessary EMS transport, more than 70% of those with noncooperation to ED treatment had alcohol ingestion [15]. In the present study, about 16% of patients in the consecutive group refused to receive ED treatment, discharged with their own will regardless of medical advice, or escaped from ED without notifying the ED personnel. In this study, many of the EMS transport for patients without cooperation to ED treatment might be unnecessary, however, even in the case of non-cooperation, the possibility of a patient requiring medical attention must be considered. Alcohol ingestion is known to be associated with frequent use of EMS transport [5]. With increasing substance abuse including alcohol abuse, the number of ED visits using EMS transport is also increasing [17]. Similar to previous studies, the present study also showed higher percentage of alcohol ingestion in the consecutive group (38.1%) than that in the non-consecutive group (13.8%).

Frequent ED visit is associated with part-time employment, retirement, and unemployed status [23]. Frequent ED visit is also related to low income status [14, 24]. The odds ratio of unemployed state for consecutive and frequent use of EMS transport was 2.8 comparing to employed state in the present study. This might be related to a high ratio of old aged patients with medical aid. It is well known that old aged people and patients with medical aid are more likely to be unemployed due to retirement, their health status, and other causes [25, 26].

More patients in the consecutive group had previous experience of EMS transport to the study hospital ED than those in the non-consecutive group in the present study. This might have led to the more use of EMS transport without hesitation in patients with previous experience of EMS transport than those without such experience. Previous EMS users might have difficulty to move due to old age, chronic diseases, or other disability [11]. If their condition is worsened during the study period, they might use EMS transport more consecutively [16, 27]. Mental illness is known to be associated with frequent EMS users [12, 28]. However, there was no significant difference in past medical or psychiatric history between the two groups in the present study. Mortality was higher in the consecutive group than that in the non-consecutive group in this study, similar to results of previous studies showing high mortality of frequent ED users [29–31]. Patients in the consecutive group might use EMS transport more frequently and consecutively due to disease progression before death.

Patients who were hospitalized after receiving ED treatment can be determined as having been involved in instances of necessary EMS transport. In a previous study, only 49% of patients transported by EMS were hospitalized, and more than 30% of patients discharged from the ED were judged to not require ED treatment [32]. In this study, 26% of patients were hospitalized for necessary EMS transport. Although a case can be determined as necessary EMS transport even if the patients are discharged (i.e., not hospitalized) with timely

appropriate ED treatment, the present study did not make any determinations regarding necessary EMS transport among discharged patients.

This study has several limitations. This retrospective study was conducted in a single university hospital located in a highly industrialized city. Thus, results of this study might not be generalizable to other rural areas or urban areas with different characteristics. It is important to obtain hospital data in EMS related studies. However, we could not obtain data from other hospitals. Previous experience of EMS transport might have been underestimated because we limited such experience to the study hospital ED. This is because we could not evaluate such experience to other hospital EDs. However, we included data only from fire stations that transported most of these patients to the study hospital ED to overcome this limitation. We did not consider the cluster effect on patients. We only examined patients that frequently used EMS transport, and we did not compare non-frequent use of EMS transport. Specific data including risky alcohol use, mortality, and previous experience of EMS transport are important elements in characterizing frequent use of EMS transport; however, these data could only be obtained from hospital data or from patients with the experience of hospitalization and frequent use of EMS transport. Risky alcohol use is highly related to frequent use of EMS transport and is necessary data in this study. Due to the retrospective nature of this study, information on risky alcohol use, and consequently data on the frequent users of EMS transport could only be obtained from patients who had been hospitalized in a general ward or intensive care unit. On the other hand, non-frequent users of EMS transport data could not be obtained since many of them were not hospitalized, so it was not included in this study. It is necessary to consider mortality data because EMS use tends to be more frequent before death. This is why the present study did not conduct a comparison between non-frequent EMS users and frequent EMS users. The comparison of less and more frequent users of EMS transport among frequent EMS users could represent a similar aspect to comparing non-frequent EMS users and frequent EMS users, from this point of view, this study would be meaningful. However, a comparative study with non-frequent users of EMS transport including risky alcohol use information will be needed in the future. Although the study period was 5 years, it is possible that the analysis period for each study subject was 1 to 4 years instead of 5 years depending on death or the time of first enrollment. Therefore, there is a possibility that consecutive groups were included less than the actual number and a longer study period would be necessary to overcome this limitation. Moreover, because the current study did not analyze patients transferred from the dominant fire station to other hospitals, it is possible that this study included fewer total patients and consecutive groups than there were in actuality.

## Conclusion

Risky alcohol use, unemployed state, and previous experience of EMS transport were associated with consecutive and frequent use of EMS transport in frequent users of EMS transport. It is necessary to reduce unnecessary use of EMS transport in frequent users of EMS through accurate evaluation of the need for EMS transport to maintain appropriate EMS system. It is possible that many frequent users of EMS transport are unnecessary users of EMS transport. In the future, it is necessary to set a definition of unnecessary use of EMS transport at the community level, and to conduct research to determine how much unnecessary use of EMS transport exists among frequent users of EMS transport. Such approaches would help inform other attempts to reduce unnecessary use of EMS transport.

## Author Contributions

**Conceptualization:** Sun Hyu Kim.

**Data curation:** Sun Hyu Kim, Hyeji Lee.

**Formal analysis:** Sun Hyu Kim.

**Investigation:** Sun Hyu Kim, Hyeji Lee.

**Methodology:** Sun Hyu Kim.

**Project administration:** Sun Hyu Kim.

**Resources:** Sun Hyu Kim, Hyeji Lee.

**Software:** Sun Hyu Kim.

**Supervision:** Sun Hyu Kim.

**Validation:** Sun Hyu Kim.

**Visualization:** Sun Hyu Kim.

**Writing – original draft:** Sun Hyu Kim.

**Writing – review & editing:** Sun Hyu Kim.

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
