## [Decision Letter · Decision Letter 0]

15 Dec 2023

PONE-D-23-23875Characteristics of Patients with Consecutive and Frequent Use of Emergency Medical Services Transport to Emergency DepartmentPLOS ONE

Dear Dr. Kim,

Thank you for submitting your manuscript to PLOS ONE. After careful consideration, we feel that it has merit but does not fully meet PLOS ONE’s publication criteria as it currently stands. Therefore, we invite you to submit a revised version of the manuscript that addresses the points raised during the review process.

We look forward to receiving your revised manuscript.

Kind regards,

Fadwa Alhalaiqa

Academic Editor

PLOS ONE

Journal Requirements:

Did you know that depositing data in a repository is associated with up to a 25% citation advantage (https://doi.org/10.1371/journal.pone.0230416)? If you’ve not already done so, consider depositing your raw data in a repository to ensure your work is read, appreciated and cited by the largest possible audience. You’ll also earn an Accessible Data icon on your published paper if you deposit your data in any participating repository (https://plos.org/open-science/open-data/#accessible-data).

4. We note that Figure 1 in your submission contain map images which may be copyrighted. All PLOS content is published under the Creative Commons Attribution License (CC BY 4.0), which means that the manuscript, images, and Supporting Information files will be freely available online, and any third party is permitted to access, download, copy, distribute, and use these materials in any way, even commercially, with proper attribution. For these reasons, we cannot publish previously copyrighted maps or satellite images created using proprietary data, such as Google software (Google Maps, Street View, and Earth). For more information, see our copyright guidelines: http://journals.plos.org/plosone/s/licenses-and-copyright.

(a) You may seek permission from the original copyright holder of Figure 1 to publish the content specifically under the CC BY 4.0 license.  

(b) If you are unable to obtain permission from the original copyright holder to publish these figures under the CC BY 4.0 license or if the copyright holder’s requirements are incompatible with the CC BY 4.0 license, please either i) remove the figure or ii) supply a replacement figure that complies with the CC BY 4.0 license. Please check copyright information on all replacement figures and update the figure caption with source information. If applicable, please specify in the figure caption text when a figure is similar but not identical to the original image and is therefore for illustrative purposes only.

**Additional Editor Comments:**

 Be sure to:Reorganize the introduction to be clearThe method used in this study including database did not fit the main purpose. Authors have to reconsider the aim of the study, method, and best database.Why you compared consecutive and non-consecutive EMS using patients rather than comparing frequent and non frequent ones? those patients might be transferred to other hospitals during study period so this information should be added to limitations.It should be better to give information about clinical status of those patients during ED admission and give ratios about how many of those transports were necessary.Give details about what do you mean by unnecessary EMS transport

Reviewers' comments:

Reviewer's Responses to Questions

**Comments to the Author**

1. Is the manuscript technically sound, and do the data support the conclusions?

Reviewer #1: No

Reviewer #2: Yes

2. Has the statistical analysis been performed appropriately and rigorously? 

Reviewer #1: No

Reviewer #2: Yes

3. Have the authors made all data underlying the findings in their manuscript fully available?

Reviewer #1: No

Reviewer #2: Yes

4. Is the manuscript presented in an intelligible fashion and written in standard English?

Reviewer #1: Yes

Reviewer #2: Yes

5. Review Comments to the Author

Reviewer #1: I thank for editors to review this paper entitled “Characteristics of Patients with Consecutive and Frequent Use of Emergency Medical Services Transport to Emergency Department” in PLOS ONE.

Author described the characteristics of frequent EMS transport and addressed the risk factors for frequent EMS use.

This paper is well-written, however, some concerns are remained before accepting.

Major revision

1 Authors used frequent EMS transport user and frequent ED user. This reviewer is confused with “the term” EMS transport user or ED user author used in text. This study examined the characteristics of frequent ED user based on the key hospital’s medical data base, not the frequent EMS user. So, the title and text should be changed along with the main theme. If author would like to examine the characteristics of frequent EMS transport user, fire department-based data base is appropriate. Do not mix up frequent EMS transport and ED visit.

2 In introduction, authors described “Users of EMS transport and …increasing. Those who … of EMS transport” in second paragraph, but “However, characteristics of … unclear” in third paragraph. These sentences say opposite meaning. Also, third paragraph lacks reasonable explanation why this study needs.

3 This reviewer did not find the reason why authors divided frequent EMS transport user into consecutive group and non-consecutive group. What is the difference between two groups from the view of the clinical settings? How important is only one year and more than one year clinically?

4 What is the definition of unnecessary EMS transport? How did authors judge this?

5 As above mentioned, authors described “to make EMS more efficient, it is important to reduce unnecessary EMS transport” in abstract, but author focused on the number of frequent ED visit. The method used in this study including database did not fit the main purpose. Authors have to reconsider the aim of the study, method, and best database.

6 In conclusion “Providing education … might be alternative solutions” no description to reach this conclusion was found in discussion.

7 In multivariable regression analysis, authors divided into cases depends on previous experience of EMS transport, therefore, this variable should not have been applied into analysis.

Minor revision

P4 Add the reference “Users of EMS transport and visitors of ED are increasing”

Add the each p-value in symptoms of patients in Table3

P9 “The percentage of patients … ED was 60%, in the consecutive…” remove comma.

Add the reason why authors classified seasons.

Add the median age (IQR) in table.

Reviewer #2: Dear authors,

This is a well written study but I advise some minor revisions. At introduction part "International review board approval" written twice. Why you compared consecutive and non-consecutive EMS using patients rather than comparing frequent and non freguent ones? those patients might be transferred to other hospitals during study period so this information should be added to limitations. You said we shouldn't decide if those transfers were necessary or not. It should be better to give information about clinical status of those patients during ED admission ang give ratios about how many of those transports were necessary. Your final sentence of discussion is incomplete. It should be better to simplfy your tables

Best regards

6. PLOS authors have the option to publish the peer review history of their article (what does this mean?). If published, this will include your full peer review and any attached files.

Reviewer #1: **Yes: **Takeshi Nishimura

Reviewer #2: **Yes: **GÜLŞAH ÇIKRIKÇI IŞIK

---

## [Author Response · Author response to Decision Letter 0]

23 Jan 2024

Thank you for your valuable time and comments. These comments have significantly improved the quality of our manuscript. We have revised this manuscript according to your comments and suggestions. 

Reviewer #1: I thank for editors to review this paper entitled “Characteristics of Patients with Consecutive and Frequent Use of Emergency Medical Services Transport to Emergency Department” in PLOS ONE.

Author described the characteristics of frequent EMS transport and addressed the risk factors for frequent EMS use.

This paper is well-written, however, some concerns are remained before accepting.

Major revision

1 Authors used frequent EMS transport user and frequent ED user. This reviewer is confused with “the term” EMS transport user or ED user author used in text. This study examined the characteristics of frequent ED user based on the key hospital’s medical data base, not the frequent EMS user. So, the title and text should be changed along with the main theme. If author would like to examine the characteristics of frequent EMS transport user, fire department-based data base is appropriate. Do not mix up frequent EMS transport and ED visit.

▶ Thank you for your comment. The study subjects were only drawn from patients of one study hospital ED identified from a hospital medical database. The relevant fire department-based database cannot be used to identify frequent EMS transport because of its anonymization and lack of specific data. Frequent ED users can refer to all patients presented to the ED by all routes, including walking, transferred from other routes, EMS transport, etc. However, the subjects included in this study were frequent ED users that were only transported to the study hospital using EMS transport. Therefore, frequent ED users in this study did not have the same meaning as frequent users of EMS transport. 

Based on this, we have revised the title as follows. 

Before

Characteristics of Patients with Consecutive and Frequent Use of Emergency Medical Services Transport to Emergency Department

After

Characteristics of Consecutive versus Non-consecutive Frequent Emergency Medical Services Transport to a Single Emergency Department 

The definition of the frequent use of EMS transport has previously been described in the method section as follows. 

The definition of frequent use of EMS transport was arbitrary following previous studies [14,15]. In this study, frequent calls for EMS transport were considered if patients called for EMS transport for more than five times per year [7]. However, not all patients who called for EMS transport were transferred to ED. In Korea, it has been reported that 67 % of patients calling for EMS transport are transferred to ED [3]. Therefore, we defined frequent users of EMS transport as those who visited ED using EMS transport for more than three times per year. To evaluate characteristics of frequent users of EMS transport, we classified frequent users of EMS transport into two groups: 1) those who used EMS transport frequently for only one year (non-consecutive group); and 2) those who used EMS transport frequently for more than two years consecutively (consecutive group). Consecutive group were defined as patients who visited ED using EMS transport more than three times for one year and more than five times for two years consecutively [16].

2 In introduction, authors described “Users of EMS transport and …increasing. Those who … of EMS transport” in second paragraph, but “However, characteristics of … unclear” in third paragraph. These sentences say opposite meaning. Also, third paragraph lacks reasonable explanation why this study needs.

▶ Thank you for your comment. We have revised the third paragraph of the introduction to read as follows. 

Before

To make EMS more efficient, it is important to reduce unnecessary EMS transport. However, characteristics of frequent users of EMS transport are currently unclear. Therefore, the objective of this study was to examine characteristics concerning frequent users of EMS transport by comparing patients who used EMS transport frequently for one year and those who used EMS transport for more than two years consecutively. 

After

Cases involving uncooperative patients, the denial of ED treatment after EMS transport, the absence of any specific symptoms at ED arrival, and repetitive use with the same symptoms might all be defined as instances of unnecessary EMS transport, along with other cases [15]. Although the judgment of unnecessary EMS transport might later turn out to be erroneous, it is important to reduce unnecessary EMS transport to make EMS more efficient. To this end, it is important to conduct accurate analysis of frequent use of EMS transport, which is likely to account for a large portion of unnecessary EMS transport. Frequent users of EMS transport have been studied based on EMS department data, not hospital data in the previous studies [11,12,16,17]. Using data based on EMS department data without hospital data was limited in its ability to identify the characteristics of frequent users of EMS transport in further detail. Moreover, within frequent users of EMS transport, the characteristics of less or more frequent users of EMS transport might differ. Therefore, the objective of this study was to examine characteristics concerning frequent users of EMS transport by comparing patients who used EMS transport frequently for one year and those who used EMS transport for more than two years consecutively. 

11. Edwards MJ, Bassett G, Sinden L, Fothergill RT. Frequent callers to the ambulance service: patient profiling and impact of case management on patient utilisation of the ambulance service. Emerg Med J. 2015;32(5): 392-396. doi: 10.1136/emermed-2013-203496.

12. Knowlton A, Weir BW, Hughes BS, Southerland RJ, Schultz CW, Sarpatwari R, et al. Patient demographic and health factors associated with frequent use of emergency medical services in a midsized city. Academic emergency medicine : official journal of the Society for Academic Emergency Medicine. 2013;20(11): 1101-1111. doi: 10.1111/acem.12253.

15. Van Dillen C, Kim SH. Unnecessary emergency medical services transport associated with alcohol intoxication. J Int Med Res. 2018;46(1): 33-43. doi: 10.1177/0300060517718116.

16. Norman C, Mello M, Choi B. Identifying frequent users of an urban emergency medical service using descriptive statistics and regression analyses. The western journal of emergency medicine. 2016;17(1): 39-45. doi: 10.5811/westjem.2015.10.28508.

17. Evans CS, Platts-Mills TF, Fernandez AR, Grover JM, Cabanas JG, Patel MD, et al. Repeated emergency medical services use by older adults: analysis of a comprehensive statewide database. Annals of emergency medicine. 2017;70(4): 506-515.e503. doi: 10.1016/j.annemergmed.2017.03.058.

3 This reviewer did not find the reason why authors divided frequent EMS transport user into consecutive group and non-consecutive group. What is the difference between two groups from the view of the clinical settings? How important is only one year and more than one year clinically?

▶ Thank you for your careful review. 

Comparing between frequent and non-frequent users may be the best way to characterize frequent EMS users. Although several studies have been conducted using EMS department-based data, specific data such as mortality, risky alcohol use, and previous experience of EMS transport, all of which are important data for identifying frequent EMS users, cannot be obtained from EMS department-based data, as they are instead obtained from hospital data. Such data can also be obtained from hospitalized patients and by scrutinizing frequent use of EMS transport. It is also necessary to consider mortality data because EMS use tends to be more frequent before death. Risky alcohol use is also an important factor that is likely to be highly related to frequent EMS use. In particular, it is very important to consider hospital data that includes data on risky alcohol use; however, accurate information on risky alcohol use could only be obtained for hospitalized patients, due to the characteristics of the hospital data. In the case of the research hospital, all daily drinking habits are recorded in detail as nursing information for inpatients only, and there were few such data for patients in the emergency room medical records only. Therefore, although most of the frequent EMS users have records of hospitalization and information on risky alcohol use could be obtained, many of the non-frequent EMS users only had emergency room medical records and had not been hospitalized, thus making it difficult to obtain comprehensive information on risky alcohol use. Previous experience of EMS transport is important data for characterizing frequent EMS use. Therefore, we aimed to analyze only frequent EMS users by comparing less and more frequent EMS users at the planning stage of this study, and a consecutive frequent group was defined as the more frequent EMS users group. This is why the present study did not conduct a comparison between non-frequent EMS users and frequent EMS users. To overcome this limitation, the author intended to indirectly compare non-frequent EMS users and frequent EMS users by conducting a comparison between less and more frequent users of EMS transport within frequent EMS users. Here, less frequent users of EMS transport refers to non-consecutive frequent users of EMS transport whereas more frequent users of EMS transport refers to consecutive frequent users of EMS transport. 

In this context, we have revised the limitations section to read as follows. 

Before

We only examined patients with frequent use of EMS transport. Risky alcohol use is highly related to frequent use of EMS transport and is necessary data in this study. Due to the retrospective nature of this study, information on risky alcohol use, and consequently data on the frequent users of EMS transport could only be obtained from patients who had been hospitalized in a general ward or intensive care unit. On the other hand, non-frequent users of EMS transport data could not be obtained since many of them were not hospitalized, so it was not included in this study. The comparison of less and more frequent users of EMS transport among frequent EMS users could represent a similar aspect to comparing non-frequent EMS users and frequent EMS users, from this point of view, this study would be meaningful. However, a comparative study with non-frequent users of EMS transport including risky alcohol use information will be needed in the future.

After

We only examined patients that frequently used EMS transport, and we did not compare non-frequent use of EMS transport. Specific data including risky alcohol use, mortality, and previous experience of EMS transport are important elements in characterizing frequent use of EMS transport; however, these data could only be obtained from hospital data or from patients with the experience of hospitalization and frequent use of EMS transport. Risky alcohol use is highly related to frequent use of EMS transport and is necessary data in this study. Due to the retrospective nature of this study, information on risky alcohol use, and consequently data on the frequent users of EMS transport could only be obtained from patients who had been hospitalized in a general ward or intensive care unit. On the other hand, non-frequent users of EMS transport data could not be obtained since many of them were not hospitalized, so it was not included in this study. It is necessary to consider mortality data because EMS use tends to be more frequent before death. This is why the present study did not conduct a comparison between non-frequent EMS users and frequent EMS users. The comparison of less and more frequent users of EMS transport among frequent EMS users could represent a similar aspect to comparing non-frequent EMS users and frequent EMS users, from this point of view, this study would be meaningful. However, a comparative study with non-frequent users of EMS transport including risky alcohol use information will be needed in the future.

The definition of a non-consecutive group has previously been described in further detail in the methods section as follows.

While there is a possibility of being classified into the consecutive group depending on the extension of the study period or the time of death, frequent users of EMS transport for only one year during the study period were defined as a non-consecutive group. For example, to be classified cases of the frequent user of EMS transport for the first time in the last year of the study period, death in the first year of the study period, or moving to a distant area from the study hospital in the first year of the study period were defined as a non-consecutive group even if there was a possibility of being classified into the consecutive group if the study period was extended or the time of death was after the second year of the study. 

It should also be noted that the limitation regarding the definition of the non-consecutive group has been described as follows in the limitation section. 

Although the study period was 5 years, it is possible that the analysis period for each study subject was 1 to 4 years instead of 5 years depending on death or the time of first enrollment. Therefore, there is a possibility that consecutive groups were included less than the actual number and a longer study period would be necessary to overcome this limitation. 

4 What is the definition of unnecessary EMS transport? How did authors judge this?

▶ The authors defined unnecessary EMS transport in cases where the patient did not receive ED treatment after EMS transport, as was done in a previous study. (Van Dillen C, Kim SH. Unnecessary emergency medical services transport associated with alcohol intoxication. J Int Med Res. 2018;46(1): 33-43). The cause of unnecessary EMS transport were as patient-oriented (against medical advice) and physician-oriented (based on doctors' suggestions) causes. Details of patient-oriented causes were classified into denial of treatment after EMS use, uncooperative towards ED healthcare providers, desire for transfer to another hospital and desire for treatment at outpatient department (OPD). Details of physician oriented causes were classified as no symptoms at ED arrival, alcohol intoxicated state without medical problem or trauma, patients can be treated in outpatient department and other hospital treatment and repeat visit to study hospital with the same symptoms. Of course, even if a case is initially judged as being an instance of unnecessary EMS transport, there may be such cases where treatment is indeed needed later, so the judgment for unnecessary EMS transport is likely to change over time. Cases of patient-oriented causes, including uncooperative patient and denial of ED treatment after EMS transport, and physician-oriented causes, including no specific symptoms at ED arrival, might also be defined as unnecessary EMS transport. 

The following statements have been reflected in changes in the introduction.

Cases involving uncooperative patients, the denial of ED treatment after EMS transport, the absence of any specific symptoms at ED arrival, and repetitive use with the same symptoms might all be defined as instances of unnecessary EMS transport, along with other cases [15]. Although the judgment of unnecessary EMS transport might later turn out to be erroneous, it is important to reduce unnecessary EMS transport to make EMS more efficient. To this end, it is important to conduct accurate analysis of frequent use of EMS transport, which is likely to account for a large portion of unnecessary EMS transport.

5 As above mentioned, authors described “to make EMS more efficient, it is important to reduce unnecessary EMS transport” in abstract, but author focused on the number of frequent ED visit. The method used in this study including database did not fit the main purpose. Authors have to reconsider the aim of the study, method, and best database.

▶ We have revised the objective and conclusion to read as follows. 

Before, objective

To make EMS more efficient, it is important to reduce unnecessary EMS transport. However, characteristics of f

---

## [Decision Letter · Decision Letter 1]

14 Mar 2024

Characteristics of Consecutive versus Non-consecutive Frequent Emergency Medical Services Transport to a Single Emergency Department

PONE-D-23-23875R1

Dear Dr. Sun Hyu Kim,

We’re pleased to inform you that your manuscript has been judged scientifically suitable for publication and will be formally accepted for publication once it meets all outstanding technical requirements.

Kind regards,

Fadwa Alhalaiqa

Academic Editor

PLOS ONE

Additional Editor Comments (optional):

Reviewers' comments:

Reviewer's Responses to Questions

**Comments to the Author**

1. If the authors have adequately addressed your comments raised in a previous round of review and you feel that this manuscript is now acceptable for publication, you may indicate that here to bypass the “Comments to the Author” section, enter your conflict of interest statement in the “Confidential to Editor” section, and submit your "Accept" recommendation.

Reviewer #2: All comments have been addressed

2. Is the manuscript technically sound, and do the data support the conclusions?

Reviewer #2: Yes

3. Has the statistical analysis been performed appropriately and rigorously? 

Reviewer #2: Yes

4. Have the authors made all data underlying the findings in their manuscript fully available?

Reviewer #2: Yes

5. Is the manuscript presented in an intelligible fashion and written in standard English?

Reviewer #2: Yes

6. Review Comments to the Author

Reviewer #2: Required revisions were done and final version of the paper is appropriate for publication. Best regards

7. PLOS authors have the option to publish the peer review history of their article (what does this mean?). If published, this will include your full peer review and any attached files.

Reviewer #2: No

---

## [Editor Report · Acceptance letter]

29 Apr 2024

PONE-D-23-23875R1 

PLOS ONE

Dear Dr. Kim, 

I'm pleased to inform you that your manuscript has been deemed suitable for publication in PLOS ONE. Congratulations! Your manuscript is now being handed over to our production team.

Kind regards, 

on behalf of

Pro Fadwa Alhalaiqa 

Academic Editor

PLOS ONE